

# A large scale evaluation of TBProfiler and Mykrobe for antibiotic resistance prediction in *Mycobacterium tuberculosis*

Pierre Mahé, Meriem El Azami, Philippine Barlas and Maud Tournoud

Data Analytics Department, bioMérieux, Marcy l'Etoile, France

## ABSTRACT

Recent years saw a growing interest in predicting antibiotic resistance from whole-genome sequencing data, with promising results obtained for *Staphylococcus aureus* and *Mycobacterium tuberculosis*. In this work, we gathered 6,574 sequencing read datasets of *M. tuberculosis* public genomes with associated antibiotic resistance profiles for both first and second-line antibiotics. We performed a systematic evaluation of `TBProfiler` and `Mykrobe`, two widely recognized softwares allowing to predict resistance in *M. tuberculosis*. The size of the dataset allowed us to obtain confident estimations of their overall predictive performance, to assess precisely the individual predictive power of the markers they rely on, and to study in addition how these softwares behave across the major *M. tuberculosis* lineages. While this study confirmed the overall good performance of these tools, it revealed that an important fraction of the catalog of mutations they embed is of limited predictive power. It also revealed that these tools offer different sensitivity/specificity trade-offs, which is mainly due to the different sets of mutation they embed but also to their underlying genotyping pipelines. More importantly, it showed that their level of predictive performance varies greatly across lineages for some antibiotics, therefore suggesting that the predictions made by these softwares should be deemed more or less confident depending on the lineage inferred and the predictive performance of the marker(s) actually detected. Finally, we evaluated the relevance of machine learning approaches operating from the set of markers detected by these softwares and show that they present an attractive alternative strategy, allowing to reach better performance for several drugs while significantly reducing the number of candidate mutations to consider.

# INTRODUCTION

With more than a million deaths and 10 million new cases worldwide in 2017 (*World Health Organization, 2018a*), tuberculosis, the disease caused by *Mycobacterium tuberculosis*, is a major public health concern. According to the World Health Organization (WHO), "urgent action is required to improve the coverage and quality of diagnosis." Indeed, due to the slow growing nature of this microorganism, the time needed to obtain a complete antibiotic resistance profile using conventional culture-based approaches can take up to several weeks, which hampers the timely prescription of an

Corresponding author
Pierre Mahé,
pierre.mahe@biomerieux.com

optimal treatment as well as patient medication adherence (*World Health Organization, 2018a*). Molecular PCR-based approaches endorsed by WHO offer faster diagnostics solutions, but currently cover a limited number of drugs, which is not sufficient to determine an optimal therapy for multi-drug resistant pathogens (*Lange et al., 2018*). Moreover, these assays target a small number of candidate mutations, making unreliable the prediction of susceptibility from a negative result (*Sanchez-Padilla et al., 2015*). By assaying the entire genome, whole-genome sequencing (WGS) holds promise to cope with this limitation, and recent years saw a growing interest in predicting antibiotic resistance of a *M. tuberculosis* strain from its WGS. In a recent technical guide (*World Health Organization, 2018b*), WHO considers NGS as "invaluable" to guide diagnosis and treatment, but also for early detection of M/XDR-*M. tuberculosis* outbreaks. Accordingly, several softwares were recently developed to do so (*Coll et al., 2015*; *Bradley et al., 2015*; *Feuerriegel et al., 2015*; *Steiner et al., 2014*; *Kohl et al., 2018*). They typically work by detecting pre-defined mutations, mostly single-nucleotide polymorphisms (SNPs), within the sequencing data, either reads or an assembled genome, and call the strain resistant whenever one of these mutations is detected. Several recent studies (*Schleusener et al., 2017*; *Macedo et al., 2018*; *Phelan et al., 2016*; *Kohl et al., 2018*) benchmarked some of these tools on limited panels of genomes (e.g., 91, 54, and 10 for the four aforementioned studies). Predictive performance estimated from such small sample sizes is however subject to a large uncertainty. The results obtained across studies are hard to reconcile, which hampers making recommendations regarding their use in clinical routine. For instance, the sensitivity of `TBProfiler` for predicting resistance to streptomycin was estimated as 57% by *Schleusener et al. (2017)* and *Kohl et al. (2018)*, while *Macedo et al. (2018)* reported 95%. In this work, we aimed to evaluate the predictive power of `TBProfiler` (*Coll et al., 2015*) and `Mykrobe` (*Bradley et al., 2015*), two widely recognized softwares allowing to predict resistance in *M. tuberculosis* for both first and second-line antibiotics, on a very large database of genomes. For this purpose, we gathered 6,574 publicly available genomes and analyzed in depth the results obtained by these tools in terms of their overall predictive performance, their ability to detect common makers and the consistency of lineages inferred. In addition, we assessed the individual predictive power of the markers they embed, as advocated by *Miotto et al. (2017)*, suggested earlier by *Farhat et al. (2016)*, and highly recommended by *World Health Organization (2018b)*. Finally, the size of the genome database allowed us to study how these software performances vary across the major *M. tuberculosis* lineages.

The so-called "direct association" strategy (*Yang et al., 2017*) that is used by `TBProfiler` and `Mykrobe`, consisting in calling a strain resistant whenever a candidate marker is detected, makes it difficult to finely control their trade-off in terms of sensitivity and specificity. Indeed, while sensitivity mechanistically increases with the number of markers considered, markers with limited predictive power can be detrimental in terms of specificity. Like *Yang et al. (2017)* and *Farhat et al. (2016)*, but on a much larger dataset, we finally evaluated whether multivariate supervised learning models can exploit the set of markers detected within a genome in a better way than direct-association approaches. We demonstrate that they can be an attractive alternative strategy, offering
in particular the possibility to finely control the trade-off between sensitivity and specificity. They also present the advantage of relying on a limited number of mutations, each having a different weight in the final decision rule, reflecting the different individual power of each marker. Finally, additional candidate mutations identified as a by-product of the `TBProfiler` genotyping process were also considered, leading to an increased performance for several drugs.

## MATERIALS AND METHODS

### Dataset constitution

A dataset of 6,616 genomes with reads available on the sequence read archive (SRA) and resistance phenotypes was gathered from several studies (*Coll et al., 2018*; *Bradley et al., 2015*; *Schleusener et al., 2017*; *Vincent et al., 2012*) and from the ReSeqTB website (https://platform.reseqtb.org/), as of June 2017. After removal of duplicated genomes, for which the raw reads had the same nucleotidic content, the final dataset included 6,574 genomes. All sequencing experiments were based on the Illumina technology and their great majority involved Illumina HiSeq or Miseq instruments.

The resulting dataset contains a variable number of phenotypes for 20 antibiotics. The current study focused on the 10 antibiotics addressed by `TBProfiler` and/or `Mykrobe`, namely amikacin, capreomycin, ethambutol, ethionamide, fluoroquinolones, isoniazid, kanamycin, pyrazinamide, rifampicin, and streptomycin. Although addressed by `TBProfiler`, para aminosalicylic acid was not considered due to the limited number of strains with phenotypes available (45 resistant for 368 susceptible ones). Moreover, since `TBProfiler` and `Mykrobe` both provide a prediction for the entire family of fluoroquinolones, a composite phenotype was created by aggregating the phenotypes of ofloxacin, moxifloxacin and ciprofloxacin: it was set as the corresponding phenotype when they agreed, and set as missing when they did not. For each antibiotic, the number of strains ranges from 635 (ethionamide) to 6,464 (isoniazid), with a small fraction of resistant strains ranging from 14.7% to 37.8% (see Table 1 for further details).

Command-line versions of `TBProfiler` (version 0.3.4) and `Mykrobe` (v0.3.3-0-gc211bf2) were run on these 6,574 samples, using their default configurations (see Supplementary Materials for further details). A resistance genotype matrix, encoding the presence/absence of all the mutations detected in the entire panel, was then built for each software. In total, 469 distinct mutations were detected by `TBProfiler` out of the 1,195 candidate mutations embedded in its catalog. In contrast, `Mykrobe`, which considers a smaller catalog detected 272 distinct mutations. Table 1 compares, for each software and each antibiotic, the number of detected and candidate markers. The difference can be important in some cases. Given the large number of samples considered in this study, the predictive power of the undetected mutations is most likely to be limited.

We noted that `TBProfiler` provides the frequency with which a given marker was detected, defined as the fraction of reads presenting the resistance marker among reads mapped at this position. A similar information can be computed from the results of `Mykrobe`, which reports the number of hits obtained for a given marker and its corresponding reference allele. This frequency information allowed us to investigate

**Table 1  Dataset constitution.**

| | Number of strains | | | | TBProfiler markers | | | Mykrobe markers | |
|---|---|---|---|---|---|---|---|---|---|
| | **Total** | **S** | **R** | **%R** | **Found** | **Candidates** | **Novel** | **Found** | **Candidates** |
| Amikacin | 1,478 | 1,110 | 368 | 24.9 | 9 | 10 | 1,588 | 7 | 9 (3) |
| Capreomycin | 1,432 | 1,086 | 346 | 24.2 | 9 | 29 | 2,005 | 9 | 12 (6) |
| Ethambutol | 5,193 | 4,432 | 761 | 14.7 | 72 | 183 | 6,552 | 24 | 30 (12) |
| Ethionamide | 635 | 395 | 240 | 37.8 | 23 | 44 | 1,859 | – | – |
| Fluoroquinolones | 1,606 | 1,250 | 356 | 22.2 | 33 | 47 | 2,970 | 32 | 191 (11) |
| Isoniazid | 6,464 | 4,770 | 1,694 | 26.2 | 100 | 309 | 3,027 | 44 | 126 (46) |
| Kanamycin | 1,154 | 815 | 339 | 29.4 | 11 | 14 | 1,588 | 8 | 15 (5) |
| Pyrazinamide | 1,188 | 841 | 347 | 29.2 | 154 | 315 | 1,562 | 56 | 163 (71) |
| Rifampicin | 6,425 | 5,188 | 1,237 | 19.3 | 53 | 132 | 4,640 | 55 | 544 (40) |
| Streptomycin | 3,506 | 2,490 | 1,016 | 29.0 | 27 | 43 | 2,305 | 51 | 156 (56) |

**Note:**
Number of strains (total, susceptible, and resistant). Number of markers detected in at least one strain ("found") by TBProfiler and Mykrobe among their entire catalogs ("candidates"). The number of novel mutations, obtained as a by-product of TBProfiler genotyping process is reported in the "novel" column. Note that some mutations are associated to several drugs, hence summing the number of mutations found in this table exceeds the figures mentioned in the main text. Note also that the TBProfiler mutations were defined at the proteic level (i.e., a single candidate mutation was counted when several nucleotidic mutations lead to the same proteic mutation). The numbers of candidate mutations considered by Mykrobe shown between brackets were taken directly from the catalog embedded in the software. Since many mutations are defined in terms of genomic or proteomic positions instead of specific alternative alleles, these numbers were translated into theoretical number of alternative alleles by multiplying genomic and proteomic positions by 3 and 19, respectively.

whether considering a minimum frequency threshold to call a marker present could be beneficial in terms of prediction or not.

Interestingly, as a by-product of its genotyping process, TBProfiler detects novel mutations within the 33 resistance loci it considers, with respect to their reference sequence in the H37Rv genome. As can be seen from Table 1, the number of these novel mutations can be relatively high (22,883 in total, but more than 50% observed in a single genome). While not used to predict resistance, their predictive power can be empirically measured, as will be described later on.

As an ending remark, we note that TBProfiler and Mykrobe did not provide results for three and two samples, respectively.

## Multivariate modeling

Motivated by the observation that many markers embedded in TBProfiler and Mykrobe catalogs seem to individually have a limited predictive power, we aimed to evaluate whether alternative multivariate machine learning approaches operating from the set of markers detected by these softwares could improve over the direct-association strategy. For this purpose, we represented each genome by a vector of binary variables encoding the presence or absence of the mutations detected by each tool on the entire panel of genomes, and relied on the Lasso-penalized logistic regression to learn a resistance prediction model for each antibiotic. The logistic regression model is appealing in this context since it provides a probabilistic prediction, thereby allowing to measure the confidence of the prediction, and to control the trade-off between sensitivity and specificity by adjusting its decision threshold. Coupled with the Lasso penalty, it leads to sparse solutions allowing to identify key (combinations of) mutations, hence to interpretable models.

To estimate the predictive performance of the Lasso-penalized logistic regression model, we relied on a nested 10-fold cross-validation procedure, that is, a 10-fold cross-validation procedure with an inner optimization of the regularization parameter, based on the AUC criterion. The final models were similarly obtained by applying a standard cross-validation procedure, designed to maximize the AUC, on the entire dataset. To test whether a difference observed in terms of AUC was significantly different from zero, we relied on the non-parametric approach proposed by *DeLong, DeLong & Clarke-Pearson (1988)*, which is implemented in the R package pROC (*Robin et al., 2011*). In particular, the statistical significance was assessed at the 0.01 level.

## Data availability

All data used and obtained in this study are available as Supplementary Materials. They contain in particular, (i) a table containing the reference phenotypes and the SRA accessions, (ii) the resistance genotype matrices obtained by `TBProfiler` and `Mykrobe`, (iii) the inferred lineages, (iv) the predictive power of individual markers, and (v) the final models by the Lasso-penalized logistic regression models (i.e., mutations selected and beta coefficients in the corresponding multivariate logistic model). Note that accessing and using data originating from the ReseqTB platform is subject to the following terms and conditions: https://platform.reseqtb.org/main/acceptTerms.html.

# RESULTS AND DISCUSSION

## Inferred lineages

A very high level of consistency between lineages inferred by both softwares was observed, consistently with a previous study (*Schleusener et al., 2017*). We noted however that `TBProfiler` provided "mixed" lineage calls for 228 samples out of the 6,571 successfully processed. These ambiguities were especially observed among the West-African, bovis and bovis/African lineages, for which `TBProfiler` systematically provided mixed results. Still, between 1.8% and 5.8% of ambiguous results were observed among the four major lineages (lineages 1–4). Further analyzes revealed however that in many cases one of the lineages was called from a marker detected at a low frequency, and that including a minimum frequency threshold to detect a lineage allowed to significantly decrease the number of ambiguous results observed among main lineages. The above rates decreased indeed respectively to 0.5–2% and 0.1–0.8% when a minimum threshold of 0.1 and 0.2 was considered. Among the 6,342 genomes unambiguously classified by both softwares, they disagreed on 21 samples only, among which 19 were called "unknown" by one software or the other. Moreover, more than 99% of the samples originate from four main lineages: lineages 1 (East-African/Indian Ocean—9.5%), 2 (Beijing/East-Asia—15%), 3 (Delhi/Central-Asia—16%), and mainly 4 (European/American—59%). This is illustrated in Figs. S1–S3.

## Resistance genotyping agreement

To assess the agreement of the softwares in terms of resistance, we first evaluated their agreement at the marker level, that is, their ability to detect the same markers within the

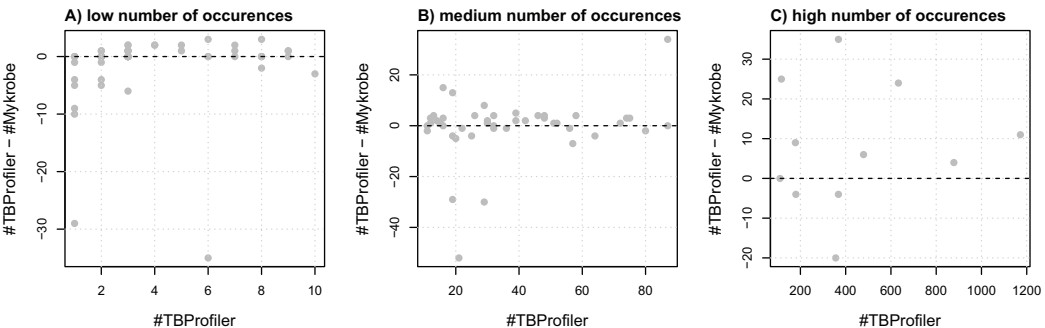

**Figure 1 Comparison of the number of calls made for the 116 markers addressed by both TBProfiler and Mykrobe.** Each dot corresponds to a marker and shows the difference in the number of calls made by TBProfiler and Mykrobe vs. the number of calls made by TBProfiler. To improve readability, markers are split in three groups, whether they are found in fewer than 10 strains (A), between 10 and 100 (B) or more than 100 strains (C) by TBProfiler. No minimum frequency threshold is considered to call a marker present.

same panel of genomes. For this purpose, we extracted a list of 116 mutations embedded in both catalogs and compared the number of calls made by both softwares. We noted that the number of calls differed quite significantly in some cases, as illustrated in Fig. 1. For instance, three markers detected in a single strain by TBProfiler are detected in 10 strains or more by Mykrobe, with one being detected in almost 30 strains. Likewise, a marker detected in around 20 strains by TBProfiler is detected in more than 70 strains by Mykrobe. Conversely, however, a marker detected in around 400 strains by TBProfiler is detected in around 35 strains less by Mykrobe. It is worth noting that these results were obtained without considering any minimum frequency threshold for calling a marker present. Similar results were also obtained when considering a threshold of 0.5 (see Fig. S4), indicating that the observed discrepancies did not result from calling present markers observed at a low frequency. This analysis therefore revealed that the discrepancies in R/S predictions made by both softwares may not only be due to differences in the catalogs of mutations they embed, but also to their underlying genotyping pipelines.

We then compared their overall agreement in terms of R/S prediction. For this purpose, we worked from the 6,570 samples and the nine antibiotics for which both tools provided a prediction (Mykrobe does not predict resistance to ethionamide). Overall, TBProfiler and Mykrobe agreed on 95.4% of the 28,426 reference phenotypes available, with 92.4% of agreement for the 6,451 resistant reference phenotypes, and 94.1% for the 21,975 susceptible ones.

## Predictive performance

Table 2 gives a summary of TBProfiler and Mykrobe predictive performance, measured in terms of the following indicators:

- *sensitivity*: the fraction of resistant strains predicted as such (also called *recall*).
- *specificity*: the fraction of susceptible strains predicted as such.

- *precision*: the fraction of resistant strains among strains predicted as resistant (also called *positive predictive value*).

We first noted that performance differed between both tools, `TBProfiler` offering in general a higher sensitivity for a lower specificity. This is mainly due to the fact that `TBProfiler` contains more resistance-defining mutations than `Mykrobe`, and that the prediction strategy amounts to calling a strain resistant whenever one of them is detected. Considering the average of sensitivity and specificity, referred to in the following as *macro-accuracy*, `TBProfiler` achieves a better performance than `Mykrobe` for five drugs (capreomycin, ethambutol, kanamycin, pyrazinamide, and fluoroquinolones). Interestingly however, `Mykrobe` outperformed `TBProfiler` in terms both of sensitivity and specificity for streptomycin (by 4.9 points of macro-accuracy), the sole antibiotic for which it detected more mutations than `TBProfiler`, especially within the *gid* gene (41 mutations instead of 1).

We also noted that the performance varied across antibiotics. Sensitivity was only around 80% for streptomycin and capreomycin, and even lower for pyrazinamide (59.1% and 34.4% with `TBProfiler` and `Mykrobe`, respectively). Specificity generally exceeded 90%, but precision was quite low in some cases (e.g., around 57% and 68% using `TBProfiler` for ethionamide and ethambutol, respectively). For ethambutol and `TBProfiler`, for instance, this means that while 92.5% of the R strains were indeed identified as such, only 67.8% of the strains predicted to be resistant were actually resistant. This might seem contradictory with the fact that the specificity was quite high (92.5%, meaning that 92.5% of the S strains were recognized as such), but was simply a consequence of the fact that the dataset involved around five times more S than R strains. While the precision (or positive predictive value) is an indicator of great clinical interest, since it is helpful to interpret the results actually provided by the software, it intrinsically depends on the prevalence of R strains in the panel studied hence must be interpreted with care (*Zignol et al., 2018*).

Strikingly, we observed major differences between our performance estimations and that reported in previous studies. For instance, streptomycin sensitivities differed by more than 20 points from the ones reported in *Schleusener et al. (2017)*, 57% for both `TBProfiler` and `Mykrobe`, instead of 77.4% and 81.4%, respectively, in the current study. Likewise, performance differed by more than 10 points for isoniazid (84% specificity for `TBProfiler` in *Schleusener et al. (2017)* instead of 97%, and 79% sensitivity for `Mykrobe` instead of 88.5%). These differences can be explained by the small sample size, 91 strains considered in *Schleusener et al. (2017)*, leading to uncertain performance prediction estimations.

We noted finally that the minimum frequency threshold considered to call a marker present sometimes had an impact on the predictive performance. This was especially the case with `TBProfiler` for ethambutol, pyrazinamide and rifampicin, where an improvement of up to three points of macro-accuracy was obtained for rifampicin, as shown in Fig. S5. `Mykrobe`, on the other hand, showed a more stable behavior, with marginal improvements obtained for streptomycin and pyrazinamide. Accordingly, the

**Table 2 Overall performance of `TBProfiler` and `Mykrobe` measured in terms of sensitivity (sensi), specificity (speci), precision, and macro-accuracy (macro), defined as the average between sensitivity and specificity.**

| | TBProfiler | | | | | Mykrobe | | | | |
|---|---|---|---|---|---|---|---|---|---|---|
| | Sensi | Speci | Precision | Macro | Thresh | Sensi | Speci | Precision | Macro | Thresh |
| Amikacin | 92.1 (89.3–94.9) | 87.9 (86–89.8) | 71.7 (67.6–75.8) | 90.0 | 0.00 | 82.6 (78.7–86.5) | 98.5 (97.8–99.2) | 94.7 (92.2–97.2) | 90.5 | 0.00 |
| Capreomycin | 80.9 (76.8–85) | 95.9 (94.7–97.1) | 86.4 (82.7–90.1) | 88.4 | 0.10 | 78.3 (74–82.6) | 94 (92.6–95.4) | 80.7 (76.5–84.9) | 86.2 | 0.00 |
| Ethambutol | 92.5 (90.6–94.4) | 92.5 (91.7–93.3) | 67.8 (65–70.6) | 92.5 | 0.25 | 87.5 (85.1–89.9) | 93.7 (93–94.4) | 70.1 (67.2–73) | 90.6 | 0.35 |
| Ethionamide | 85.3 (80.8–89.8) | 61.5 (56.7–66.3) | 57.2 (52.1–62.3) | 73.4 | 0.10 | – | – | – | – | – |
| Fluoroquinolones | 89 (85.7–92.3) | 95.8 (94.7–96.9) | 85.9 (82.3–89.5) | 92.4 | 0.00 | 85.1 (81.4–88.8) | 97.2 (96.3–98.1) | 89.6 (86.3–92.9) | 91.2 | 0.00 |
| Isoniazid | 89.3 (87.8–90.8) | 97 (96.5–97.5) | 91.3 (89.9–92.7) | 93.2 | 0.30 | 88.5 (87–90) | 98.3 (97.9–98.7) | 94.8 (93.7–95.9) | 93.4 | 0.00 |
| Kanamycin | 91.7 (88.8–94.6) | 95.7 (94.3–97.1) | 89.9 (86.7–93.1) | 93.7 | 0.00 | 81.7 (77.6–85.8) | 98 (97–99) | 94.5 (91.9–97.1) | 89.8 | 0.00 |
| Pyrazinamide | 59.1 (53.9–64.3) | 97.1 (96–98.2) | 89.5 (85.5–93.5) | 78.1 | 0.25 | 34.4 (29.4–39.4) | 99.2 (98.6–99.8) | 94.4 (90.4–98.4) | 66.8 | 0.35 |
| Rifampicin | 91.4 (89.8–93) | 98.3 (97.9–98.7) | 92.6 (91.1–94.1) | 94.8 | 0.20 | 92.4 (90.9–93.9) | 98.3 (97.9–98.7) | 92.8 (91.4–94.2) | 95.3 | 0.00 |
| Streptomycin | 77.4 (74.8–80) | 91 (89.9–92.1) | 77.7 (75.1–80.3) | 84.2 | 0.30 | 81.4 (79–83.8) | 96.7 (96–97.4) | 91.1 (89.2–93) | 89.1 | 0.40 |

**Note:**
For each software and antibiotic, the minimum frequency threshold considered to call a marker present was chosen to maximize the macro-accuracy. 95% confidence intervals are provided for sensitivity, specificity, and precision.

results presented in Table 2 were obtained by optimizing this minimum frequency threshold, for each software and each antibiotic, to maximize the corresponding macro-accuracy. An identical table was included in Table S1 when no such minimum frequency thresholds were considered. In practice, we recommend to use the thresholds provided in Table 2.

## Predictive performance by lineage

The size of the dataset considered allowed then to evaluate how `TBProfiler` and `Mykrobe` behaved across lineages. For this purpose, we focused on the four main lineages, that is lineages 1–4, and computed the same performance indicators per lineage. Table 3 summarizes the results obtained for `TBProfiler`. The global entries correspond to the performance obtained from the 6,342 samples coming from these four major lineages, and excluding all samples with ambiguous lineage calls. These values can slightly differ from the values in Table 2, obtained from the entire dataset of 6,571 samples, including samples of other lineages and ambiguous lineage calls. As before, the minimum frequency threshold to call a marker present was fine-tuned, for each drug, to maximize the macro-accuracy indicator on the entire dataset (i.e., of the global entry).

We noted a large discrepancy across lineages for some antibiotics. This was notably the case of capreomycin, ethambutol, ethionamide, pyrazinamide, and streptomycin, where a difference of more than 10 points of macro-accuracy was observed between the lineages of highest and lowest performance. This difference reached 46.4 points for capreomycin and 34.1 points for pyrazinamide, where no resistant strain from lineage 1 was detected in both cases, and exceeded 17 points for ethionamide and streptomycin. Interestingly, performance observed within lineage 4 was not systematically better than within other lineages, despite the fact that it accounted for more than 50% of the samples for all drugs but ethionamide, where it represented 44% of the samples. We also noted

**Table 3 TBProfiler performance across the four major lineages.**

| Drug | Lineage | Total | S | R | %R | Sensitivity | Specificity | Macro | Precision |
|---|---|---|---|---|---|---|---|---|---|
| Amikacin | Global | 1,440 | 1,074 | 366 | 25 | 92.1 (±2.8) | 87.5 (±2) | 89.8 | 71.5 (±4.8) |
| | Lineage1 | 24 | 19 | 5 | 21 | 60 (±42.9) | 68.4 (±20.9) | 64.2 | 33.3 (±53.3) |
| | Lineage2 | 600 | 359 | 241 | 40 | 96.7 (±2.3) | 84.7 (±3.7) | 90.7 | 80.9 (±5) |
| | Lineage3 | 75 | 46 | 29 | 39 | 93.1 (±9.2) | 91.3 (±8.1) | 92.2 | 87.1 (±12.6) |
| | Lineage4 | 741 | 650 | 91 | 12 | 81.3 (±8) | 89.4 (±2.4) | 85.3 | 51.7 (±11.4) |
| Capreomycin | Global | 1,395 | 1,052 | 343 | 25 | 80.8 (±4.2) | 96.2 (±1.2) | 88.5 | 87.4 (±3.9) |
| | Lineage1 | 23 | 22 | 1 | 4 | 0 (±0) | 90.9 (±12) | 45.5 | 0 (±0) |
| | Lineage2 | 569 | 348 | 221 | 39 | 89.6 (±4) | 94.3 (±2.4) | 91.9 | 90.8 (±4) |
| | Lineage3 | 71 | 60 | 11 | 15 | 90.9 (±17) | 73.3 (±11.2) | 82.1 | 38.5 (±30.2) |
| | Lineage4 | 732 | 622 | 110 | 15 | 62.7 (±9) | 99.7 (±0.4) | 81.2 | 97.2 (±3.9) |
| Ethambutol | Global | 5,011 | 4,260 | 751 | 15 | 92.8 (±1.8) | 92.3 (±0.8) | 92.5 | 68.1 (±3.5) |
| | Lineage1 | 412 | 395 | 17 | 4 | 94.1 (±11.2) | 96.7 (±1.8) | 95.4 | 55.2 (±24.4) |
| | Lineage2 | 870 | 440 | 430 | 49 | 97.4 (±1.5) | 70 (±4.3) | 83.7 | 76 (±4.1) |
| | Lineage3 | 876 | 840 | 36 | 4 | 91.7 (±9) | 95.7 (±1.4) | 93.7 | 47.8 (±17) |
| | Lineage4 | 2,853 | 2,585 | 268 | 9 | 85.4 (±4.2) | 94.4 (±0.9) | 89.9 | 61.2 (±6.3) |
| Ethionamide | Global | 628 | 392 | 236 | 38 | 85.6 (±4.5) | 61.5 (±4.8) | 73.5 | 57.2 (±6.8) |
| | Lineage1 | 6 | 5 | 1 | 17 | 100 (±0) | 100 (±0) | 100 | 100 (±0) |
| | Lineage2 | 312 | 206 | 106 | 34 | 84 (±7) | 44.2 (±6.8) | 64.1 | 43.6 (±10.3) |
| | Lineage3 | 32 | 30 | 2 | 6 | 100 (±0) | 86.7 (±12.2) | 93.3 | 33.3 (±65.3) |
| | Lineage4 | 278 | 151 | 127 | 46 | 86.6 (±5.9) | 78.8 (±6.5) | 82.7 | 77.5 (±7.8) |
| Fluoroquinolones | Global | 1,555 | 1,204 | 351 | 23 | 89.2 (±3.2) | 95.7 (±1.1) | 92.5 | 85.8 (±3.9) |
| | Lineage1 | 72 | 66 | 6 | 8 | 50 (±40) | 100 (±0) | 75 | 100 (±0) |
| | Lineage2 | 411 | 255 | 156 | 38 | 93.6 (±3.8) | 89.8 (±3.7) | 91.7 | 84.9 (±5.8) |
| | Lineage3 | 161 | 129 | 32 | 20 | 93.8 (±8.4) | 98.4 (±2.2) | 96.1 | 93.8 (±8.6) |
| | Lineage4 | 911 | 754 | 157 | 17 | 85.4 (±5.5) | 96.8 (±1.3) | 91.1 | 84.8 (±6.1) |
| Isoniazid | Global | 6,218 | 4,558 | 1,660 | 27 | 89.7 (±1.5) | 96.9 (±0.5) | 93.3 | 91.3 (±1.4) |
| | Lineage1 | 596 | 502 | 94 | 16 | 94.7 (±4.5) | 88.8 (±2.8) | 91.8 | 61.4 (±10.1) |
| | Lineage2 | 944 | 325 | 619 | 66 | 95.6 (±1.6) | 95.4 (±2.3) | 95.5 | 97.5 (±1.3) |
| | Lineage3 | 1,009 | 842 | 167 | 17 | 88.6 (±4.8) | 99.2 (±0.6) | 93.9 | 95.5 (±3.3) |
| | Lineage4 | 3,669 | 2,889 | 780 | 21 | 84.6 (±2.5) | 97.8 (±0.5) | 91.2 | 91.3 (±2.2) |
| Kanamycin | Global | 1,117 | 781 | 336 | 30 | 92 (±2.9) | 95.5 (±1.5) | 93.8 | 89.8 (±3.4) |
| | Lineage1 | 25 | 19 | 6 | 24 | 50 (±40) | 94.7 (±10.1) | 72.3 | 75 (±49) |
| | Lineage2 | 383 | 189 | 194 | 51 | 95.9 (±2.8) | 93.7 (±3.5) | 94.8 | 93.9 (±3.4) |
| | Lineage3 | 69 | 41 | 28 | 41 | 92.9 (±9.5) | 100 (±0) | 96.5 | 100 (±0) |
| | Lineage4 | 640 | 532 | 108 | 17 | 87 (±6.3) | 95.9 (±1.7) | 91.5 | 81 (±7.9) |
| Pyrazinamide | Global | 1,136 | 796 | 340 | 30 | 59.7 (±5.2) | 97 (±1.2) | 78.3 | 89.4 (±4.2) |
| | Lineage1 | 124 | 117 | 7 | 6 | 0 (±0) | 100 (±0) | 50 | 0 (±0) |
| | Lineage2 | 244 | 81 | 163 | 67 | 52.8 (±7.7) | 97.5 (±3.4) | 75.2 | 97.7 (±3.2) |
| | Lineage3 | 141 | 110 | 31 | 22 | 51.6 (±17.6) | 100 (±0) | 75.8 | 100 (±0) |
| | Lineage4 | 627 | 488 | 139 | 22 | 72.7 (±7.4) | 95.5 (±1.8) | 84.1 | 82.1 (±7.5) |

(Continued)

| Drug | Lineage | Total | S | R | %R | Sensitivity | Specificity | Macro | Precision |
|---|---|---|---|---|---|---|---|---|---|
| Rifampicin | Global | 6,181 | 4,957 | 1,224 | 20 | 91.7 (±1.5) | 98.3 (±0.4) | 95 | 93 (±1.5) |
| | Lineage1 | 597 | 570 | 27 | 5 | 92.6 (±9.9) | 97.9 (±1.2) | 95.2 | 67.6 (±18.3) |
| | Lineage2 | 929 | 333 | 596 | 64 | 96.8 (±1.4) | 96.1 (±2.1) | 96.4 | 97.8 (±1.2) |
| | Lineage3 | 1,008 | 928 | 80 | 8 | 85 (±7.8) | 99.1 (±0.6) | 92 | 89.5 (±7.3) |
| | Lineage4 | 3,647 | 3,126 | 521 | 14 | 86.8 (±2.9) | 98.3 (±0.5) | 92.5 | 89.7 (±2.8) |
| Streptomycin | Global | 3,381 | 2,381 | 1,000 | 30 | 77.9 (±2.6) | 90.2 (±1.2) | 84.1 | 77 (±3) |
| | Lineage1 | 222 | 198 | 24 | 11 | 70.8 (±18.2) | 92.4 (±3.7) | 81.6 | 53.1 (±23.7) |
| | Lineage2 | 766 | 256 | 510 | 67 | 98.6 (±1) | 80.9 (±4.8) | 89.8 | 91.1 (±2.5) |
| | Lineage3 | 284 | 240 | 44 | 15 | 65.9 (±14) | 96.7 (±2.3) | 81.3 | 78.4 (±15) |
| | Lineage4 | 2,109 | 1,687 | 422 | 20 | 54.5 (±4.8) | 90.5 (±1.4) | 72.5 | 58.8 (±6.4) |

**Note:**
Figures between brackets correspond to 95% confidence intervals. Shown in gray are the lineages with less than 100 strains. Shown in orange and green are the lineages where the macro accuracy is lesser or greater than the global one by more than five points. These thresholds were set arbitrarily.

that a lower performance could be due either to a lack of sensitivity, as it was the case for amikacin/lineage 4, capreomycin/lineage 4 and streptomycin/lineage 4, or to a lack of specificity, for ethambutol/lineage 2 and ethionamide/lineage 2. Similar observations could be made from the results obtained by Mykrobe, shown in Table S2.

### Individual markers performance

As mentioned above TBProfiler and Mykrobe consider a pre-defined list of resistance-defining mutations and predict a strain resistant whenever one of these mutations is detected. The number and the quality of these markers therefore intrinsically define the overall performance of the softwares. Analyzing individually the predictive power of the markers is interesting in at least two respects. First, to get a better understanding of the discrepancies observed previously in terms of the overall predictive performances of the softwares. Moreover, as described by Miotto et al. (2017) and suggested earlier by Farhat et al. (2016), to appreciate the prediction provided by the algorithm: the prediction may be deemed more confident when it is based on a marker with a higher predictive power.

We first noted that a limited number of markers had a high level of sensitivity. Considering all drugs, 80.6% and 70.6% of the markers detected by TBProfiler and Mykrobe, respectively, had a sensitivity smaller than 1%. Only 19 and 14 of TBProfiler and Mykrobe markers had a sensitivity above 10%, and only seven above 40% in both cases. In contrast, the great majority of markers were highly specific: 98.6% of the TBProfiler and all Mykrobe markers had a specificity greater than 95%. This is illustrated in Fig. S6.

Figure 2 shows the individual markers performance obtained for amikacin and streptomycin to illustrate how they can help better understanding the trade-offs achieved by both softwares. For amikacin, we noted that the predictive power was mostly driven by a specific mutation in both cases. Table S3 indicates that it corresponded to the mutation rss_1401A > G, which was captured by both softwares, albeit in a different number of

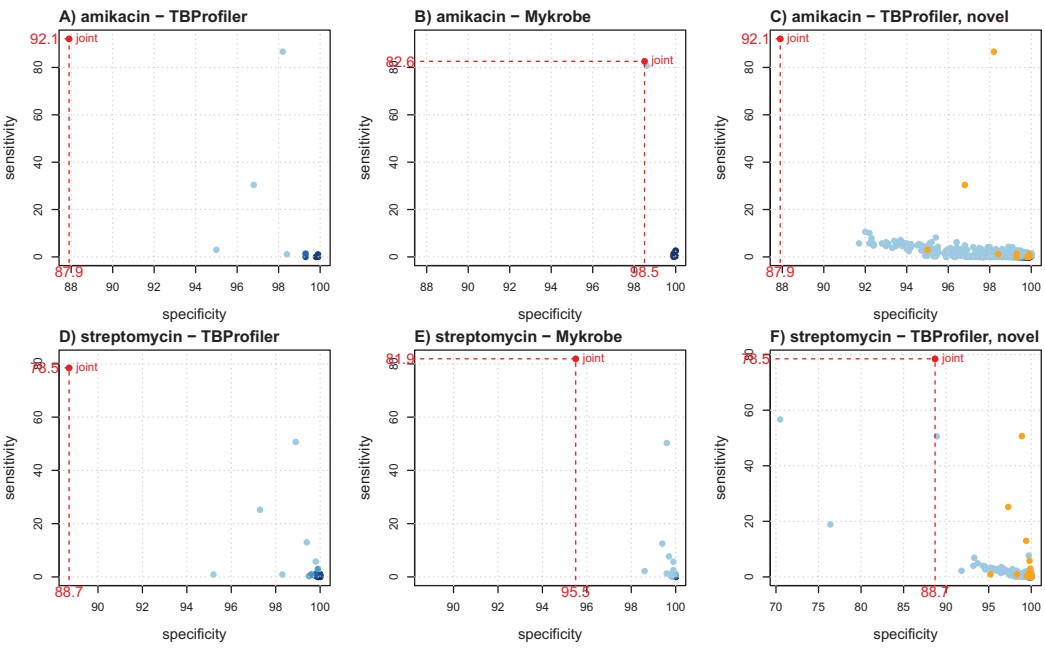

**Figure 2 Illustration of individual markers performance for amikacin (A–C) and streptomycin (D–F), in terms of sensitivity and specificity.** (A and D) `TBProfiler` markers; (B and E) `Mykrobe` markers; (C and F) novel mutations identified by `TBProfiler` (in blue), together with the original mutations (orange). The values shown in red correspond to the overall performance of the tools. No minimum frequency threshold is considered to call a marker present.

strains (339 for `TBProfiler` and 311 for `Mykrobe`). This mutation taken individually would lead to a sensitivity of 86.7% and a specificity of 98.2% for `TBProfiler`, and of 80.7% and 98.6% for `Mykrobe`. All the other mutations considered by `Mykrobe` are highly specific but not very sensitive (2.7% at best, see Table S3), hence only allowing for a marginal improvement over the sensitivity brought by the rss_1401A > G mutation. `TBProfiler`, on the other hand, includes a mutation with an intermediate level of sensitivity, not belonging to the `Mykrobe` catalog. It corresponds to a mutation at position 514 in *rss*, and while it indeed allows improving the overall sensitivity of `TBProfiler`, it has a specificity lower than any mutation considered by `Mykrobe` for amikacin (96.8% instead of 98.6%, see Table S3). Including this mutation improves `TBProfiler` sensitivity but at the cost of a reduced specificity. Choosing to include or not this mutation amounts to favoring sensitivity over specificity, and is therefore a matter of choice regarding the target performance one wants to achieve. In the case of streptomycin, `Mykrobe` is both more sensitive and specific than `TBProfiler`. Interestingly, Fig. 2 suggested a similar behavior to that of amikacin, `TBProfiler` including a marker of intermediate sensitivity and lesser specificity. While this can explain the lower specificity of `TBProfiler`, this marker does not allow `TBProfiler` to outperform `Mykrobe` in terms of sensitivity in this case. This therefore suggests that the `Mykrobe` catalog is more thorough for this antibiotic. We note from Table 1 that `Mykrobe` has detected 51 mutations for this antibiotic, instead of 27 by `TBProfiler`. It turns out that `Mykrobe` contains much

**Table 4 Performance obtained using Lasso-penalized logistic regression models operating from the markers detected by TBProfiler or Mykrobe.**

|  | TBProfiler **markers** | | Mykrobe **markers** | |
| --- | --- | --- | --- | --- |
|  | **AUC** | **Support** | **AUC** | **Support** |
| Amikacin | 92.4 | 6/9 | 89.5 | 2/7 |
| Capreomycin | 87.1 | 2/9 | 85.8 | 3/9 |
| Ethambutol | 92.4 | 21/72 | 91 | 11/24 |
| Ethionamide | 82.4 | 12/23 | – | – |
| Fluoroquinolones | 92.3 | 11/33 | 90.6 | 8/32 |
| Isoniazid | 94.2 | 18/100 | 92.7 | 7/44 |
| Kanamycin | **93.7** | 5/11 | 89.1 | 3/8 |
| Pyrazinamide | **76.1** | 60/154 | 65.3 | 20/56 |
| Rifampicin | 95 | 21/53 | 95.1 | 21/55 |
| Streptomycin | 87.7 | 9/27 | 88.3 | 45/51 |

Note:
AUC was estimated by the nested cross-validation procedure described in the Materials and Methods section. The support corresponds to the number of markers included in the model, over the number of candidate markers (see Table 1). AUC values shown in bold are significantly different between the two models. Statistical significance is assessed at the 0.01 level.

more mutations in *gid* than TBProfiler (41 instead of one). We note however, that some novel mutations detected by TBProfiler may have a good predictive power. For streptomycin, this is in particular the case of a highly specific mutation (99.7%) having a sensitivity of 7.7% which, if not already explained by other markers, may bring additional predictive power. As can be seen on Fig. 2, only three markers of the native TBProfiler catalog actually exhibit a greater sensitivity. These novel mutations are not taken into account by TBProfiler in its prediction process, but can readily be considered in multivariate machine learning algorithms, as will be described below.

A complete table providing the individual performance of the known and novel mutations is available as Supplementary Materials to allow a finer exploration.

## Multivariate modeling strategies

### Models based on the original TBProfiler and Mykrobe catalogs of markers

We first built models from the markers included in the TBProfiler and Mykrobe catalogs, separately, using the nested cross-validation procedure described above to estimate the generalization performance of the models. Note that we considered only the markers related to a given antibiotic to build the corresponding prediction model.

Table 4 summarizes the results obtained in terms of AUC and support size, defined as the number of markers involved in the final model. Performance was comparable for most antibiotics, but better AUCs were obtained by the model learned from TBProfiler markers for kanamycin and pyrazinamide (significantly at the 0.01 significance level, see Methods). Interestingly, the number of markers involved in the models was relatively limited, with often less than 20 markers. This represents a drastic reduction with respect to the size of the original catalogs. ROC curve analysis was performed to objectively compare the performance of these models and that of TBProfiler and Mykrobe.

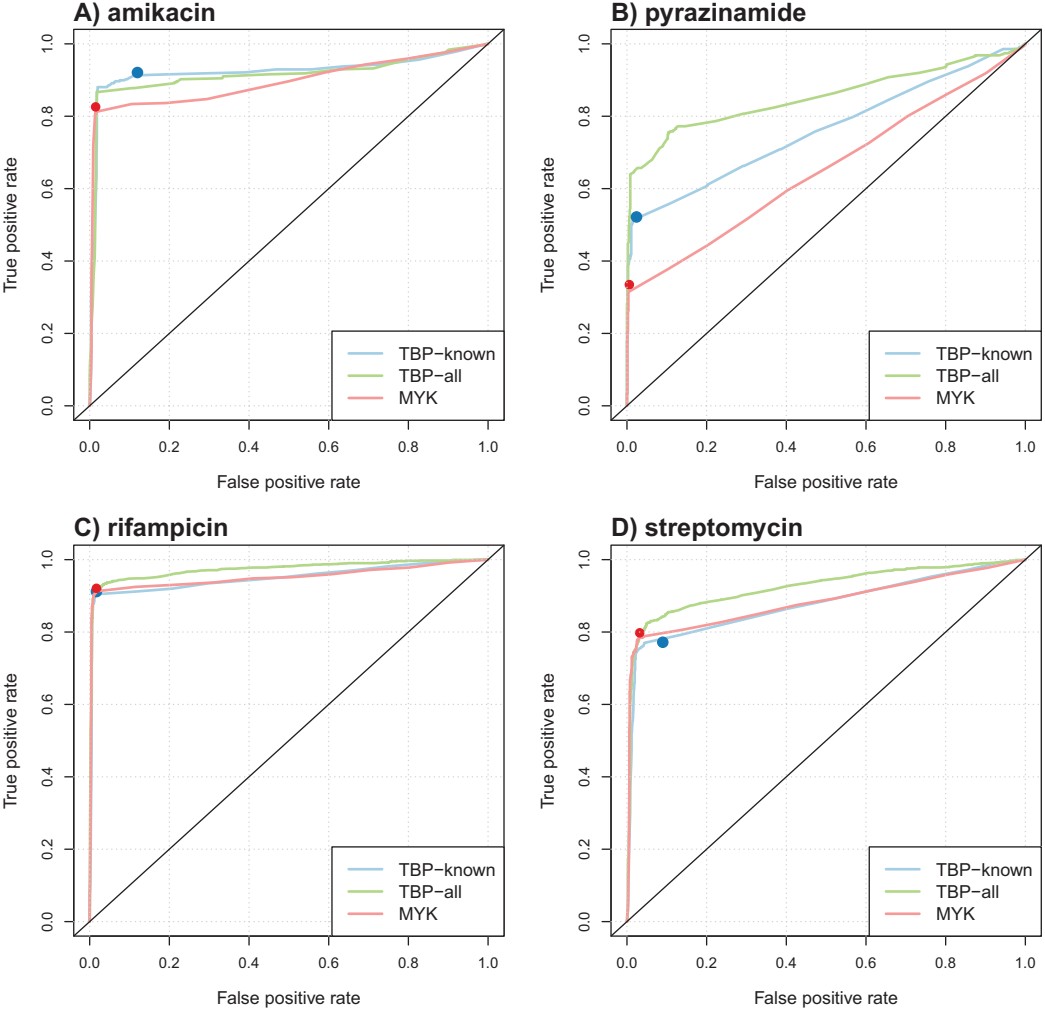

**Figure 3 Illustration of ROC curves obtained by L1-penalized logistic regression using** `TBProfiler` **and** `Mykrobe` **markers, for amikacin (A), pyrazinamide (B), rifampicin (C) and streptomycin (D).** The TBP-known model is built using the `TBProfiler` known markers only, the TBP-all model using the known and the novel mutations identified by `TBProfiler`, and the MYK model using the `Mykrobe` markers. The red and blue dots represent performances respectively obtained by `TBProfiler` and `Mykrobe` softwares under the same cross-validation process.

The resulting ROC curves for four antibiotics are illustrated in Fig. 3, the curves obtained for the remaining ones are shown in Fig. S9. For all antibiotics, the performance of `TBProfiler` and `Mykrobe` lied on the ROC curves obtained for the corresponding logistic regression models. This means that a similar sensitivity/specificity trade-off can be achieved by adjusting the threshold on the probability returned by the logistic regression model, indicating that both strategies achieve the same level of performance. We noted moreover that `TBProfiler` and `Mykrobe` performance often lied where the ROC curves were the closest to the optimal point of the ROC space (which lies at $x = 0$ and $y = 1$ and denotes perfect sensitivity and specificity). The most notable exception was observed for amikacin and `TBProfiler`, where the logistic regression model allowed to trade a bit of sensitivity for a greater gain in specificity.

**Table 5** AUC obtained by Lasso-penalized logistic regression models operating from mutations detected by `TBProfiler` or `Mykrobe`, and the lineages inferred.

| | TBProfiler markers | | | | Mykrobe markers | |
|---|---|---|---|---|---|---|
| | Known markers | Known markers + lin. | All markers | All markers + lin. | Markers | Markers + lin. |
| Amikacin | 92.4 | 92.9 | 91.4 | 91.3 | 89.5 | 91.1 |
| Capreomycin | 87.1 | 88.6 | 89.4 | 90.4 | 85.8 | 87.6 |
| Ethambutol | 92.4 | **93.9** | **94.7**[*] | **94.7**[*] | 91 | **93.2** |
| Ethionamide | 82.4 | 82.9 | 84.3 | 84.1 | – | – |
| Fluoroquinolones | 92.3 | 92.8 | 93.9 | 94 | 90.6 | 90.7 |
| Isoniazid | 94.2 | 94.7 | **96**[*] | **96.2**[*] | 92.7 | **94.2** |
| Kanamycin | 93.7 | 94.1 | 94.7 | 94.7 | 89.1 | 91 |
| Pyrazinamide | 76.1 | **85.4** | **85.3** | **85.9** | 65.3 | **78.6** |
| Rifampicin | 95 | **95.9** | **97.4**[*] | **97.3**[*] | 95.1 | **96** |
| Streptomycin | 87.7 | **89.1** | **92.4**[*] | **92.4**[*] | 88.3 | **89.6** |

**Note:**
AUC values shown in bold are significantly better than the AUC obtained with the corresponding models based on "known markers" only. For models involving "all markers" detected by `TBProfiler` (i.e., known + novel ones), starred AUC ([*]) are significantly better than that of both the corresponding models based on known markers, and known markers + lineages. No significant difference was observed between models involving "all markers + lineages" and "all markers" only. Statistical significance is assessed at the 0.01 level.

Overall, while it did not improve the performance of `TBProfiler` and `Mykrobe`, the Lasso-penalized logistic regression strategy allowed to reach the same level of performance while providing a probabilistic prediction, hence allowing to adjust, to some extent, the desired levels of sensitivity and specificity, and relying on much shorter list of mutations.

### Models integrating lineage effects and novel mutations identified by TBProfiler

We then aimed to evaluate the relevance of considering novel covariates in the model to take into account lineage effects, and to evaluate whether the novel mutations identified by `TBProfiler` during its genotyping process could bring additional predictive power. For this purpose, we applied the same procedure as above, using a one-hot encoding scheme to include the lineage information in the models: a set of five mutually exclusive binary covariates were defined to encode the presence of a given genome in one of the four major lineages (1–4) or to another one. We therefore considered two configurations using `Mykrobe` data (using lineage covariates or not in addition to the detected markers), and four using `TBProfiler` data (using lineage covariates or not in addition to known markers only, or to known and novel markers).

Table 5 summarizes the results obtained in terms of AUC. The corresponding ROC curves are shown in Fig. 3 and Fig. S9. We first noted that considering lineage variables in addition to known mutations generally allowed increasing the AUC of the models, although often marginally, except for pyranizamide (9 and 13 points in terms of AUC for `TBProfiler` and `Mykrobe`, respectively). Considering novel mutations identified by `TBProfiler` also allowed significantly improving the performance of the models based on known mutations only, with the largest improvement for pyrazinamide (+9 points AUC) and streptomycin (+5 points AUC), and to a lesser extent for isoniazid and

rifampicin (+2 points AUC). Finally, considering lineage covariates in addition to the novel mutations identified by `TBProfiler` did not lead to any significant improvement, compared to novel mutations only.

This analysis therefore indicated that the novel mutations identified by `TBProfiler` are able to capture global lineage effects, and sometimes lead to further improvements. This suggests that they either capture in a finer way the underlying population structure, or contain novel causal variants. Disentangling both possibilities would require additional analyses, and is beyond the scope of this work. A closer analysis of streptomycin and pyrazinamide models, for which the gain brought by novel mutations is the highest, revealed some interesting findings. The streptomycin model includes indeed many mutations in the *gid* gene, a gene for which the `TBProfiler` catalog includes a single mutation. The two *gid* mutations of highest weights in the model based on `TBProfiler` mutations (namely, 75P > R and 134A > E) are actually part of the `Mykrobe` catalog, and are also involved in the model based on `Mykrobe` mutations. This therefore suggests once again that the `Mykrobe` catalog is more thorough for this antibiotic. Nevertheless, several other mutations in the *gid* which are not part of the `Mykrobe` catalog also have a high weight in the model based on `TBProfiler` mutations (e.g., 76G > C), indicating that they may correspond to novel causal variants. Likewise, the pyrazinamide model involves several mutations of large weights within the *pncA* gene, which are not part of the `TBProfiler` nor `Mykrobe` catalogs. Identifying the *pncA* mutations causing resistance to pyrazinamide is still an open question (*Yadon et al., 2017*). The mutations identified in this present study may therefore provide additional empirical evidence to validate the effect of novel, putatively causal, mutations.

While it is difficult to judge the biological relevance of novel candidate markers, this study demonstrated that machine learning allows to readily integrate them to build predictive models of higher performance. The ability of the Lasso-penalized logistic regression model to identify short list of mutations allows moreover to properly interpret the prediction rules and the individual effects of the mutations involved. The lists of mutations involved in the various prediction models and their weights are provided as Supplementary Materials for a finer exploration.

## CONCLUSION

We performed a large scale evaluation of two widely recognized softwares to predict antibiotic resistance in *M. tuberculosis*. The size of the considered dataset was much larger than in prior studies done in a similar setting (*Schleusener et al., 2017*; *Macedo et al., 2018*; *Phelan et al., 2016*; *Kohl et al., 2018*), hence providing a more precise estimation of their prediction performance. This study nevertheless has several limitations. First, as opposed to the aforementioned ones, this study cannot be considered as a proper independent validation study, as some of the genomes considered have been involved in the development of these tools. However, it allows to draw general conclusions about the respective merits of each tool, as well as their current limitations, as recently described in a similar setting for *Staphylococcus aureus* (*Mason et al., 2018*). Second, as any study lead in a similar setting, this study suffers from the fact that phenotypic antimicrobial

susceptibility testing is an imperfect gold-standard (*Brennan-Krohn, Smith & Kirby, 2017*). Estimating the predictive performance of models like `TBProfiler`, `Mykrobe` or any machine-learning based model is therefore intrinsically flawed by an uncertainty in the reference phenotypes. While dedicated statistical methods allow to take into account and compensate for this uncertainty (*Rutjes et al., 2007*), they require to have some knowledge about the performance of the reference (phenotypic) test, and postulate that the reference and the new tests are independent conditionally given the condition (e.g., disease or resistance), which is hard to validate. Interestingly however, a list of high-confidence mutations was recently proposed by *Miotto et al. (2017)*. It is considered trustworthy enough by the WHO to correct phenotypes determined phenotypically: isolates harboring these mutations were systematically considered as resistant in *World Health Organization (2018a)*, even if they were identified as susceptible by phenotypic testing. In the dataset considered in this study, less than 2% of susceptible strains harbor at least one of these mutations for most drugs, except fluoroquinolones and capreomycin, where this figure rises to 3–4%, and ethionamide, where it rises to 19%, most probably due to the fact that the confidence in the underlying mutation is moderate. Further details about this analysis are provided in Supplementary Materials. Definitively claiming that these susceptible samples are false negatives is probably hazardous since (i) while deemed highly confident, these mutations may not be perfectly specific and (ii) calling these mutations present depends on the genotyping pipelines embedded in `TBProfiler` and `Mykrobe`, and in particular on the allele frequency threshold considered. We therefore decided to consider the phenotypes as defined in the original publications, but emphasize that this uncertainty in the reference phenotypes has to be kept in mind when interpreting the results of such prediction models. Finally, as it is also the case for any study led in a similar setting, the presence of groups of highly-related isolates (e.g., coming from an outbreak) may bias the predictive performance estimation. A standard way to circumvent this issue would be to identify such groups of close isolates using a SNP-based distance criterion defined at the whole-genome level, and to pick one isolate per group. This would require however to have access to the assembled genomes of the isolates, which is not provided by `TBProfiler` nor, and is beyond the scope of this work. As a first step in this direction, we nevertheless aimed to identify such groups of close isolates from the list of "novel" mutations identified by `TBProfiler` among the resistance loci considered. While probably much less sensitive than a whole-genome distance criterion and harder to interpret, this preliminary analysis, described in Supplementary Materials, indicates that close isolates are indeed probably present in the dataset (especially within the samples coming from the original `Mykrobe` study (*Bradley et al., 2015*)), but suggests that this issue is probably marginal. An interesting perspective of this work could amount to consolidating this analysis after a preliminary step of genome assembly, in order ultimately to refine the estimation of the predictive performances after the exclusion of such close isolates.

These limitations acknowledged, this large-scale study confirms the overall good performance of these tools, hence the relevance of NGS for resistance prediction in *M. tuberculosis*, in agreement with *World Health Organization (2018b)*. This study

moreover revealed that an important fraction of the catalog of mutations they embed is of limited predictive power. This was especially the case for `TBProfiler`, where around two third of the 1,195 mutations included in its catalog were actually never detected in the large dataset considered in this study (more than 6,500 genomes). It also showed that both softwares achieve different trade-offs in terms of sensitivity and specificity, which is directly related to the nature of the mutation catalogs they embed, `TBProfiler` including a larger catalog than `Mykrobe`, leading in general to a higher sensitivity for a lesser specificity, but also to the fact that they do not fully agree in terms of genotyping. Finally, this study revealed strong lineage effects for some antibiotics, with much lesser performance in some lineages than others. These results are consistent with *World Health Organization (2018b)*, who reported important differences in resistance performance prediction across countries in a multi-country surveillance project. We therefore believe that the predictive performance of these softwares should be quantified in a lineage per lineage basis, to allow the user to better appreciate the predictive value of the result obtained depending on the lineage inferred from his sample.

We demonstrate moreover that standard machine learning approaches operating from the set of markers detected by these softwares provide an interesting alternative strategy. Indeed, they seek for signatures involving the smallest number of mutations (e.g., 165 `TBProfiler` mutations altogether for the 10 antibiotics considered, out of 1,126 candidates from the original catalog), while preserving the prediction performance. Such approaches are of great importance to design PCR assays, targeting a small but optimal number of makers. The development of targeted assays for first- and second-line *M. tuberculosis* therapies, with good performance, would allow to circumvent the cost and training requirements involved by NGS, which is still an issue in low- and middle-income countries (*World Health Organization, 2018b*). Moreover, the final decision rule gives different weights to individual markers, reflecting this way the difference in individual predictive performance of markers as advocated by *World Health Organization (2018b)*. Contrary to direct-association strategies whose performance depends on the catalog of mutations they embed, probabilistic models also offer a better control of the sensitivity/ specificity trade-off. In addition, they provide a natural way to include additional covariates, as shown in this work with the novel mutations identified by `TBProfiler` during its genotyping process. Interestingly, these novel mutations allowed to improve significantly the performance within some lineages in particular, especially for streptomycin and pyrazinamide, as shown in Fig. S10, which therefore suggests that some of these mutations may be relevant.

Finally, we noted that predictive performance remains poor within specific lineages for some antibiotics (e.g., streptomycin/L4 or ethionamide/L2). This suggests that some resistance mechanisms remain to be deciphered, and that current lists of markers may be biased toward specific lineages. Agnostic *k*-mer based approaches operating from the entire genome could be relevant in reducing this selection bias (*Drouin et al., 2016*; *Davis et al., 2016*; *Mahé & Tournoud, 2018*). This dataset and study will provide a solid benchmark for evaluating more advanced machine learning strategies to predict antibiotic resistance in *M. tuberculosis* from WGS data.

## ACKNOWLEDGEMENTS

Data used in the preparation of this article were obtained from the ReSeqTB Data Platform. As such, the investigators within the organizations that contributed data to the ReSeqTB Data Platform assisted with the design and implementation of the data platform and/or provided data, but did not participate in the analysis of the data or the writing of this report.

### Funding

The authors received no funding for this work

### Competing Interests

All authors are employees of bioMérieux, a company creating and developing infectious disease diagnostics. No further potential conflicts of interest relevant to this article are reported.

### Author Contributions

- Pierre Mahé conceived and designed the experiments, performed the experiments, analyzed the data, contributed reagents/materials/analysis tools, prepared figures and/or tables, authored or reviewed drafts of the paper, approved the final draft.
- Meriem El Azami conceived and designed the experiments, performed the experiments, analyzed the data, approved the final draft.
- Philippine Barlas contributed reagents/materials/analysis tools, approved the final draft.
- Maud Tournoud conceived and designed the experiments, analyzed the data, authored or reviewed drafts of the paper, approved the final draft.

### Data Availability

All data used and obtained in this study are available as Supplementary Files.

### Supplemental Information

Supplemental information for this article can be found online at http://dx.doi.org/10.7717/peerj.6857#supplemental-information.

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
