# Peer review of "A large scale evaluation of TBProfiler and Mykrobe for antibiotic resistance prediction in Mycobacterium tuberculosis"

_PeerJ, doi:10.7717/peerj.6857_

## Round 0.1 · original submission · Major Revisions

I suggest you address all comments, particularly those of reviewer 3, in order to make a stronger argument about your results.

Reviewer 1 ·

Basic reporting

It is written in a good English wording and clear to me as reviewer.
Referencing is partially good, I have some suggestions that provided later.

Experimental design

It falls with the scope of the journal.

I found the methods fine and enough professional for conduct such a good research in this field.

Validity of the findings

I agree with the topic but not seen enough text on rationale for this research. I am not satisfied. Please include more text to convince the readers and reviewers happy with your data and findings,

More literature review in Scopus and Web of Science is necessary to finish the discussion with it.


Authors should have shown how they can use this predictive tool in prediction of antibiotic resistance in MT.
Is this new approach in agreement which WHO guideline in treatment of such problematic bacilli?

Additional comments

basically I like the MS but it can be considered after some minor changes in the revision.

·

Basic reporting

no comment

Experimental design

In the section lines 87-94, it would be helpful to know whether the TBProfiler and Mykrobe programs were run with their respective default settings, or, if different settings were attempted (apart from allele frequency cutoffs). For example, TBProfiler offers a number of different read mappers (bwa, bowtie, minimap2) and variant callers (lofreq, bcftools) which may have influence on the number and frequencies of detected resistance markers.

Validity of the findings

no comment

Reviewer 3 ·

Basic reporting

no comment

Experimental design

About methods, the Authors stated they used the command line versions of the Mikrobe and the TBprofiler pipelines. I was not able to find detailed quality thresholds, variant call thresholds, coverage thresholds and similar parameters used in the evaluation study.

Validity of the findings

no comment

Additional comments

The Authors stated despite specificity was generally high (90%), precision was found low, especially for ethambutol and ethionamide. I would suggest the Authors to reconsider this in the view of the fact that conventional phenotypic DST is an imperfect gold standard. As demonstrated by several studies indeed, (for example) liquid MGIT phenotypic DST systematically misses resistance due to specific mutations. In particular, ethambutol phenotypic DST is one of the most affected by this problem with the M306V embB mutation validated as marker of ethambutol resistance by functional genetics but testing either phenotypically susceptible by MGIT in many cases. Even worst if we consider the M306I that is causing a MIC close to the critical concentration used for phenotypic DST. This is an important point to be considered when evaluating NGS software. In this sense, discrepancies between variant calling among the two pipelines would be more relevant, compared to the performance against phenotypic DST.

The Authors found very different performance characteristics compared to previously reported studies. One of the reason could be that previous studies were performed on "controlled" samples, whereas in the current study - except for the ReSeqTB datasets that ensure a level of data curation - could not check for phenotypic DST accuracy in the metadata. I would suggest to mention this possibility among the reasons of such differences. It would be interesting have a quick insight on this by taking the list of markers of R reported in Miotto ERJ 2017 and looking how many isolates harboring such mutations have been reported as phenotypically S in the dataset.

Could the Authors exclude the presence of duplicate isolates in the dataset?

The question about the lineage-specific performance needs to be some adjustment. First, is the dataset excluding highly-related isolates (e.g. <12 SNPs of difference on a distance matrix)? If the dataset is including strains belonging to an outbreak, this could effect the analysis about the performances on different lineages. Second, it is well known that the frequency of some mutations is depending and varying upon geographical region; since lineage distribution is also geographical-dependent, a given mutation can show different sensitivity accordingly. The extreme case is the S450L in rpoB that is working very well worldwide, except in Swaziland (eSwatini), where the most (and nearly solely) rpoB mutation causing RIF-R is at codon 491. I would suggest to mention the complexity of evaluating lineage-specific performances in the manuscript. Also, whereas it is relevant to evaluate performances against different lineages, I would suggest to revise the sentence at lines 368-370 in the conclusions, as because we would need a clear frequency distribution among lineages before defining a confidence level as the one suggested. In addition, this is again depending upon the list of mutations used for interpreting sequencing data.

ReSeqTB raw sequencing data are provided according to specific terms and conditions. Please clearly state if the datasets used are publicly available and/or any specific permission has been approved.

---

## Round 0.2 · accepted · Accept

I sincerely appreciate your having taken into account the comments raised by two independent reviewers.

# Reviewer 1 ·

Basic reporting

I have already reviewed the paper and the authors made my corrections. I agree with positive decision for this paper.

Experimental design

it is fine by me

Validity of the findings

valued

Additional comments

nice job

Reviewer 3 ·

Basic reporting

no comment

Experimental design

no comment - reviewers' questions addressed

Validity of the findings

no comment

Additional comments

-